# An Efficient Trust-Aware Task Scheduling Algorithm in Cloud Computing Using Firefly Optimization

**DOI:** 10.3390/s23031384

**Published:** 2023-01-26

**Authors:** Sudheer Mangalampalli, Ganesh Reddy Karri, Ahmed A. Elngar

**Affiliations:** 1School of Computer Science and Engineering, VIT-AP University, Amaravati 522237, India; 2Faculty of Computers and Artificial Intelligence, Beni-Suef University, Beni-Suef 62511, Egypt

**Keywords:** task scheduling, trust, availability, success rate, makespan, turnaround efficiency, ACO—ant colony optimization, SLA—service level agreement, PSO—particle swarm optimization, GA—genetic algorithm

## Abstract

Task scheduling in the cloud computing paradigm poses a challenge for researchers as the workloads that come onto cloud platforms are dynamic and heterogeneous. Therefore, scheduling these heterogeneous tasks to the appropriate virtual resources is a huge challenge. The inappropriate assignment of tasks to virtual resources leads to the degradation of the quality of services and thereby leads to a violation of the SLA metrics, ultimately leading to the degradation of trust in the cloud provider by the cloud user. Therefore, to preserve trust in the cloud provider and to improve the scheduling process in the cloud paradigm, we propose an efficient task scheduling algorithm that considers the priorities of tasks as well as virtual machines, thereby scheduling tasks accurately to appropriate VMs. This scheduling algorithm is modeled using firefly optimization. The workload for this approach is considered by using fabricated datasets with different distributions and the real-time worklogs of HPC2N and NASA were considered. This algorithm was implemented by using a Cloudsim simulation environment and, finally, our proposed approach is compared over the baseline approaches of ACO, PSO, and the GA. The simulation results revealed that our proposed approach has shown a significant impact over the baseline approaches by minimizing the makespan, availability, success rate, and turnaround efficiency.

## 1. Introduction

The cloud computing paradigm provides on-demand services to users to deploy or migrate their existing on-premises infrastructure into the cloud environment, to develop cloud naïve applications in the cloud environment, or for users to make use of the existing services running in the cloud environment. This can be possible in the cloud environment by provisioning a virtual infrastructure to corresponding users with the use of virtualization technology. Users can access all types of services such as memory, computing, storage, and network on-demand whenever they need to by subscribing to those services. This subscription of services between the cloud provider and the user is named the SLA. Based on the SLA made between the user and the cloud provider, corresponding services need to be leveraged to the users. The cloud computing paradigm has characteristics [1], i.e., scalability, which renders services to its users by dynamically scaling up or down and scaling in or out based on the situation. Resource pooling is another characteristic of this paradigm which provides services to cloud users from a shared pool of resources that are, logically, synchronized with each other when a resource is allocated to a user. All these resources are given to users on demand whenever they request a cloud provider and, in turn, the cloud provider will automatically provision virtual resources to the customers based on the SLA made between them. All these services are to be billed according to their corresponding usage and pay-per-usage policy of services. Whenever a cloud user chooses services in the cloud paradigm, they are provisioned with respective services virtually from the cloud provider; in order to provide these services to customers without any issues, the cloud provider needs a scheduling mechanism that automatically assigns virtual resources to users based on their requests. Therefore, to map corresponding requests to virtual machines and assign those resources to customers, an effective task scheduler is needed in this paradigm as this paradigm is to be used by many users around the world; to assign resources to these large-scale users, an automatic provisioning mechanism should be presented in the cloud paradigm. The requests coming onto the cloud console are of variable sizes and different types that include text, images, video, streaming data, etc. These requests may come from heterogeneous resources. Therefore, to schedule these dynamic, variable, and heterogeneous requests onto precise virtual machines, an effective task scheduling algorithm is needed in this paradigm. An ineffective task scheduling mechanism leads to the degradation of the quality of the services by violating the SLAs parameters and thereby the trust between the cloud provider and the user will be degraded. Therefore, an effective task scheduling algorithm is needed for this paradigm to provide a high quality of service by maintaining the SLA and thereby improving the trust between the cloud providers and users. Many of the existing authors proposed several task scheduling mechanisms by using metaheuristic approaches, i.e., PSO [2], ACO [3], and the GA [4]. Earlier authors proposed task schedulers using metaheuristic approaches as it is the NP-hard problem to identify the best possible solution to schedule tasks to precise virtual machines. Therefore, in this research, we propose an efficient trust-aware task scheduling algorithm that calculates the priorities of dynamic-natured incoming tasks and the priorities of virtual resources based on the unit cost of electricity. This research uses the firefly optimization algorithm [5] to model scheduling in the cloud paradigm. This research mainly concentrates on the trust parameters based on the SLA metrics, i.e., the availability of virtual resources, the success rate of virtual resources, and, finally, the turnaround efficiency. The entire experiment was conducted on Cloudsim and our proposed approach, TAFFA, was evaluated against the baseline approaches for the above specified parameters. 

### Motivation and Contributions

The cloud computing paradigm provides on-demand flexible, scalable services to users based on their needs with the help of virtualization. These services automatically assign resources to customers based on their SLA made between the cloud vendor and user. To assign these virtual resources precisely based on the type of requests posed onto the cloud console, an efficient task scheduling algorithm is needed. An improper and ineffective task scheduler in the cloud paradigm degrades the quality of service by violating the SLAs parameters and thereby the degradation of trust in the cloud provider. This situation motivates us to develop a task scheduling algorithm that considers the priorities of tasks, VMs using the unit cost of electricity, and schedules tasks accordingly while minimizing the makespan and improving the availability, success rate, and turnaround efficiency. Below are the highlights and contributions of our research in this manuscript.

Developed an efficient trust-aware task scheduling algorithm using firefly optimization (TAFFA).The effective scheduling of the priorities of tasks and VMs with electricity unit costs are calculated.Identified the relation between the makespan and trust based on the SLAs parameters.The calculation of trust based on the SLA is identified, i.e., the success rate of the virtual resource, the availability of VMs, and the turnaround efficiency.A deadline constraint in used in this research in order to assign tasks to virtual resources after the current execution of the pending tasks.Extensive simulations carried on Cloudsim.Fabricated (different workload distributions, i.e., uniform, normal, left, and right skewed distributions) and realtime worklogs from HPC2N and NASA are used.The proposed TAFFA is evaluated over PSO, ACO, and the GA and finally the simulation results proposed TAFFAs improved parameters, i.e., the availability, success rate, and turnaround efficiency while minimizing the makespan.

The rest of the manuscript is organized as follows. Section 2 discusses the related works conducted by various authors, Section 3 discusses the problem definition and system architecture, Section 4 discusses the proposed efficient trust-aware task scheduling algorithm in cloud computing using firefly optimization, Section 5 discusses the simulation and results, and Section 6 discusses the conclusion and proposed future work.

## 2. Related Work

In [6], the authors proposed a task scheduling approach modeled by adaptive PSO which balances between the local and global search space by modifying the inertia weights of particles. It mainly addressed the parameters makespan, throughput, and average resource utilization. The entire simulation was conducted on Cloudsim and evaluated over the baseline approaches. The results revealed that linearly descending adaptive PSO shows an impact over the existing mechanisms for the mentioned parameters. Energy consumption is an important parameter and an important aspect in the cloud computing paradigm. For this approach, the authors in [7] proposed a scheduling mechanism by a variant of PSO. In this approach, the PSO algorithm was made as an adaptive approach by changing the inertia weight and accelerated coefficient. This approach was implemented on the Cloudsim toolkit. It was evaluated over the existing max–min and min–min algorithms. From the simulation results, it is evident that the proposed approach shows a significant improvement over the baseline algorithms for the makespan and energy consumption. The cloud computing paradigm is advantageous only when users achieve a high satisfaction when resources in the cloud paradigm are properly utilized. The authors in [8] proposed a task scheduling mechanism which is a resource and deadline aware task scheduling algorithm based on PSO. It was implemented on Cloudsim. It was compared against the existing algorithms, i.e., max–min and min–min approaches. From the results, it was proved that PSO-RADL outperforms over the existing baseline algorithms for the parameters, i.e., the makespan, resource utilization, task rejection, penalty, and total cost. In [9], the authors focused on formulating a task scheduling approach using CSO and based on the behavior of cats, it was proposed and aimed at addressing the parameters, i.e., the makespan, total power cost at datacenters, migration time, and energy consumption. The scheduling model implemented on Cloudsim and two types of workloads are used for checking the efficacy of the algorithm, i.e., the random workload and real time worklogs. It was compared over PSO and CS algorithms and the results revealed that the mentioned parameters show a significant improvement over the baseline algorithms. In [10], the authors formulated a task scheduling mechanism for the balancing of the load by hybridizing lateral wolf and PSO algorithms. The LW-PSO approach works based on the fitness value of LW. It was implemented on Cloudsim with python and evaluated over the baseline approaches and it has shown a significant improvement over these algorithms while minimizing the execution time, turnaround time, and response time. In [11], the authors formulated a scheduling mechanism which addresses the parameters of the execution time and wait time. The scheduling of tasks is to be done by using the PSO approach and also by balancing the load of tasks. The entire experiment was conducted on Gridsim and evaluated with random and realtime clusters computational workloads. It was compared over different baseline approaches and from the results, it proved that Ba-PSO outperforms the existing approaches for the specified parameters. In [12], the authors formulated a task scheduling mechanism by combining PSO and opposition-based learning. This approach accelerates the searching process of PSO by using opposition-based learning to avoid a premature convergence. It was implemented on Cloudsim and evaluated over the different variations of PSO and the proposed OPSO shows a huge impact over the existing approaches for the minimization of the makespan and energy consumption. In [13], a task scheduling mechanism formulated MVO and PSO approaches. MVO works as a global search process and PSO acts as a local search. It was implemented on the Cloudsim toolkit and compared over the existing MVO and PSO algorithms and the results revealed that EMVO shows a huge impact over the baseline algorithms by minimizing the makespan while improving the resource utilization. In [14], the task scheduling algorithm was formulated by the hybridization of the GA and FPA. It was simulated on the Cloudsim toolkit and the addressed parameters were resource utilization, the completion time, and computation cost. It was compared against the existing GA and FPA and the simulation results revealed that GA-FPA outperformed the existing GA and FPA approaches for the specified parameters. The authors in [15] designed an energy efficient task scheduling mechanism in a heterogeneous cloud environment. They used a deadline constraint to execute tasks in the specified time to put the new task into the execution mode. This approach was formulated in two stages. The first stage discusses the minimization of the execution time while scheduling tasks and the second stage discusses the reassignment of tasks based on their priorities to obtain an optimized execution time and energy consumption. It was implemented on Cloudsim and compared over the existing baseline approaches of RC-GA, AMTS, and E-PAGA and from the simulation results, it is evident that the energy consumption was greatly minimized by 5 to 20% for EPETS over the baseline algorithms. In [16], a hybrid task scheduling model was developed by the authors to address the parameters of the makespan, cost, and response time. It was developed by combining two algorithms, i.e., the GA and electro search. The entire simulations were carried out on Cloudsim. It was evaluated over the baseline algorithms, i.e., HPSOGA, ES, and ACO algorithms. From the results, it was proved that HSGA was dominant over the existing algorithms for the specified parameters. Fault tolerance can be considered as one of the important parameters in the cloud paradigm as the quality of the cloud services depends on the fault tolerance. The quality of service will be preserved when the cloud provider follows the SLA promptly. For this purpose, in [17], the authors formulated a multi-phase scheduling mechanism formulated using the GA. In the initial phase, the individual fitness value was calculated and it was considered as the local fitness value; then, in the next phase, the global fitness value was identified. After calculating both fitness values, the tasks from the different users were mapped to the virtual resources according to the fitness values optimally. The total simulation was conducted on Cloudsim and it was evaluated over the baseline GA and its variants. From the results, it was observed that MFTGA showed dominance over the existing baseline approaches for an SLA violation, the execution time, memory utilization, cost, and energy consumption. When the number of user requests increased in the cloud paradigm, the computation overhead will be increased in order to schedule tasks on to respective virtual resources. The tasks in this paradigm arise from various heterogeneous resources, therefore, the authors in [18] extracted the features of the tasks and reduced the task features using mapreduce framework. It was implemented on Cloudsim and was modeled by combining the WAO and GA. It was evaluated over the existing approaches of the GA and WAO. From the results, it showed a huge impact over the baseline approaches by minimizing the processing cost, computation cost, and processing time. In [19], the authors formulated a task scheduling model based on a consciousness of energy and the cost incurred by the cloud provider. It was modeled using a GA and was formulated in two phases. In the first phase, the task priorities were calculated, but in the second phase, these tasks were allocated to the respective processors by using the GA approach. This experiment was conducted on MATLAB. It was compared against several baseline approaches and it was finally identified that it outperforms the baseline approaches while minimizing the makespan and energy consumption. For achieving a high performance in the cloud paradigm, the authors in [20] designed a task scheduling mechanism which is a hybridized approach in which the GSA is used as a local search and the GA is used as the global search. It was implemented on Cloudsim and the workload captured in this approach is a real time workflow benchmark dataset of a heterogenous nature. GSAGA was compared over Profit-MTC and IWC algorithms. From the results, it evident that GSAGA showed a huge impact over the compared approaches while maximizing its processing capacity and minimizing the energy consumption. In [21], a hybridized task scheduling approach was formulated using the FPSO and GA. The main aim of this approach was to efficiently balance the tasks among VMs. It was implemented on Cloudsim and compared over the existing PSO and GA. The results revealed that FPSO-GA had a fuzzy nature, which proved that it outperforms the existing approaches by efficiently balancing the load among VMs. In [22], the authors formulated an efficient power-aware task scheduling algorithm developed using a modified ACO approach. This algorithm was modeled based on updating pareto by accelerating the convergence using adaptive probability distribution. All the simulations were carried out on Cloudsim and evaluated over the existing state-of-the-art approaches and, finally, from the results, it was proved that MOTS-ACO was dominant over the existing approaches while distributing tasks and utilizing power efficiently. In [23], the authors designed a hybridized task scheduling algorithm which was combined using ACO and PSO, where PSO was used for the global search and ACO was used as the local search. The simulations were carried out on Cloudsim and it was evaluated over the variants of PSO and ACO approaches. From the results, it is evident that AC-PSO showed its dominance over the existing variants by minimizing the makespan, total cost, and an increase in resource utilization. In [24], the authors designed a task scheduling approach which was designed for the minimization of the makespan and improved the performance of the cloud computing paradigm. It was modeled using a R-ACO, a reinforced approach through which ant patterns can be rewarded. The experimentation was conducted on MATLAB 2015. It was evaluated over the baseline ACO approach and identified that R-ACO minimizes the makespan by 60%. In [25], a load balancing strategy was developed based on the ACO algorithm. This algorithm aims at the minimization of the makespan and cost. Cloudsim was used as a simulation tool for this approach. It was evaluated over PSO and hybrid approaches and, from the results, Mr-LBA showed its dominance over the existing algorithms for the specified parameters. In [26], a hybridized task scheduling approach was developed and aimed to address energy consumption and resource utilization. This approach was developed using ACO and BF algorithms. It was implemented using the Cloudsim toolkit. It was compared to ACO and BF approaches and the results showed its dominance in terms of the makespan, energy consumption, and resource utilization over the existing state-of-the-art algorithms. In [27], the authors designed a task scheduling approach based on the adaptive nature of the ACO algorithm. This adaptive nature was generated by using polymorphic ants based on the pheromone updation of ants. This algorithm aimed at the minimization of the execution cost and the execution time. It was implemented on Cloudsim and evaluated over the state-of-the-art approaches. From the results, it was evident that AACO outperformed the existing algorithms for the specified parameters. In [28], the authors designed a task scheduling approach for challenges such as the workflow intensity, task heterogeneity, and task dependencies. It was modeled using the hybrid approach of HEFT-ACO. It was implemented in the AWS cloud environment and it was evaluated over the existing variants of ACO. From the results, it was shown to have a huge impact over the existing variants by minimizing the makespan and cost. In [29], the authors focused on developing a task scheduling algorithm which was meant for computational- and delay-sensitive tasks at the mobile edge cloud for resource allocation. It mainly aimed at the efficient utilization of radio and computational resources to the user’s requests. It was implemented using MATLAB. It was modeled as a joint scheduling mechanism using ACO and GA approaches; ACO was used as the local and the GA was used as the global search and it was compared against the existing GA and ACO. From the results, it is evident that it increased its capability in the efficient utilization of resources and allocated them in an optimized manner. In [30], a load balancing mechanism was developed to distribute resources effectively among VMs in the cloud paradigm. Heterogeneous users’ requests arrived at a cloud console and, in order to balance these tasks properly onto the corresponding VMs, the authors in [30] used an ACO-NN based on nearest neighbor among all the ants; the best possible ants with solutions needed to be taken into the situation, and, therefore, the scheduling problem needed to be solved. It was implemented on Cloudsim and evaluated over the baseline approaches; finally, the results showed that it efficiently balanced the load among the VMs compared with the existing approaches. In [31], the authors used DGWO to model and schedule the dependent tasks in cloud computing. The main aim of this approach was to minimize the computation and transmission costs. They used the largest order value method to model scheduler in the cloud paradigm. They used workflowsim for their experimentation and, finally, they compared DGWO over PSO and BPSO algorithms; the results showed that it had a huge impact over the baseline approaches for the mentioned parameters.

In Table 1, it can be clearly observed that many of the existing authors in task scheduling addressed various parameters, i.e., the execution time, execution cost, response time, energy consumption, resource utilization, and load balancing, of the tasks. Earlier authors failed to address the parameters related to trust in the cloud provider, which is an important aspect in the cloud paradigm. An ineffective task scheduler in the cloud paradigm leads to a poorer quality of service, which in turn leads to damaging individuals’ trust in a cloud provider. In the proposed approach, the SLA-based trust parameters are evaluated which impact the quality of service and, in turn, affect one’s trust in a cloud provider. Therefore, an efficient trust-aware task scheduling algorithm based on the SLAs parameters is proposed to preserve trust in the cloud provider. For this scheduling paradigm, the priorities of incoming tasks and VMs are considered and then the tasks are carefully scheduled onto VMs while minimizing the makespan and improving the availability, success rate, and turnaround efficiency.

## 3. Problem Definition and System Architecture

This section carefully discusses the problem definition and system architecture in this research. The below subsection discusses the problem definition.

### 3.1. Problem Definition

Consider that a set of $m_{t}=\left\{m_{1},m_{2},m_{3},\ldots .m_{t}\right\}$ tasks are mapped to $n_{v}=\left\{n_{1},n_{2},n_{3},\ldots n_{v}\right\}$ virtual resources which are residing in a set of hosts indicated as $ho_{q}=\left\{ho_{1},ho_{2},ho_{3}\ldots ho_{q}\right\}$, which are in turn residing in $d_{p}=\left\{d_{1},d_{2},d_{3},\ldots d_{p}\right\}\;$ datacenters. In this research, the task scheduling problem is defined in such a way that $m_{t}$ tasks are mapped to $n_{v}$ virtual machines, which are in turn mapped to $d_{p}$ datacenters by considering the priorities of the tasks and VMs based on the unit cost of electricity, while minimizing the makespan and improving the SLA-based trust parameters, i.e., the availability, success rate, and turnaround efficiency.

### 3.2. System Architecture

This subsection discusses the system’s architecture in a detailed manner. Figure 1 discusses the system’s architecture. Initially, various cloud users from heterogeneous resources are submitted onto the cloud console. These requests are captured by a cloud broker on behalf of all the users and are submitted to the task manager. The task manager needs to examine the requests of the users and if it is a valid request from a user as per the SLA made between the cloud provider and users, then the service request dispatcher dispatches the corresponding request to the scheduler. In our architecture, we induced a mechanism at the task manager level where the priorities of tasks are based on the length, run time, and capacity of tasks. After calculating the tasks’ priorities, the VM based on the unit cost of electricity is calculated. After calculating the priorities carefully, corresponding tasks are fed to the execution queue and then the scheduler sends tasks onto the appropriate VMs based on the consumption status of virtual resources at the resource manager level, while minimizing the makespan and improving the SLA-based trust parameters. In our research, we posed a deadline constraint where no other tasks are to be scheduled onto a VM when a task is currently being executed on a VM. After discussing the system’s architecture, we carefully modeled our scheduler using a mathematical model in this architecture. Table 2 indicates the notations used in the system’s architecture. 

Initially, in this mathematical modeling, we calculated the current workload of all the VMs. It was calculated using Equation (1).
(1)$$load_{n_{v}}=\sum load_{v}$$
where $load_{v}$ indicated the workload on all n VMs considered in our work. After the calculation of the current workload on all the VMs, we calculated the workload on all the physical hosts as all the VMs are hosted on the considered physical hosts. Therefore, the current workload on all the hosts was calculated using Equation (2).
(2)$$load_{ho_{q}}=\frac{load_{n}{}_{v}}{\;ho_{q}}$$
where $load_{ho_{q}}$ indicates the workload on all the considered hosts.

After the calculation of the workload on the VMs and hosts, to map tasks onto the VMs and the priorities of the tasks are calculated based on the length and the various processing capacities of the VMs in which the tasks are running; the priorities of the tasks depend on the length of the tasks and the processing capacities of the VMs. The processing capacity of a VM was calculated using Equation (3).
(3)$$pr_{n_{v}}=pr_{no}*pr_{MIPS}$$

After calculating the processing capacity of a VM, the processing capacity of all the VMs was calculated using Equation (4).
(4)$$ToT_{n_{v}}^{pr}=\;pr_{n}{}_{v}$$

After the calculation of the total processing capacity of all the VMs, the size of the task was calculated using Equation (5).
(5)$$m_{t}^{size}=m_{t}^{MIPS}*m_{t}^{pr}$$

After calculating the processing capacity from Equation (5), the priorities of the tasks were calculated using Equation (6).
(6)$$prio_{m_{t}}=\frac{m_{t}^{size}}{pr_{n_{v}}}$$
where $pr_{n_{v}}$ indicates the processing capacity of a VM and $m_{t}^{size}$ indicates the size of a corresponding task. At this level, we evaluated the priorities of tasks but we evaluated the priorities of VMs based on the electricity unit cost as incoming tasks onto the cloud console should be precisely mapped onto the VMs by considering the length and processing capacity of the VMs. Therefore, we calculated the priorities of the VMs based on the electricity cost using Equation (7). It is defined as the ratio of the highest electricity cost among all the datacenters to the electricity cost at that corresponding datacenter.
(7)$$prio_{n}{}_{v}=\frac{high_{ele_{cost}}^{DC}}{DC_{ele_{cost}}}$$

After the calculation of the priorities of the tasks and the VMs from Equations (6) and (7), the task manager needs to submit these priorities to the scheduler and the scheduler will generate schedules to the corresponding tasks by mapping the high-priority tasks onto the VMs with low electricity price datacenters. Every time the scheduler generates schedules based on these priorities, the scheduler needs to check with the resource manager for the status of the virtual resources to see whether they are busy or idle, i.e., the resource status.

After the calculation of these priorities, we identified the SLA-based trust parameters, i.e., the availability, success rate, and turnaround efficiency, which impact the quality of service of a cloud provider; if these SLA parameters are violated, then it has an impact on the trust in the cloud provider. 

Initially, the availability of a virtual machine is calculated and it can be defined as the ratio of the number of tasks accepted by the VM to the number of tasks submitted by the user. It is calculated using Equation (8).
(8)$$AV\left(n_{v}\right)=\frac{AV_{m}}{m_{t}}$$

Another SLA-based trust parameter which needs to be discussed is the success rate, which is defined as the ratio of successfully executed requests onto a VM to the number of submitted requests for a specified amount of time.
(9)$$SR\left(n_{v}\right)=\frac{S_{m_{t}}}{AV_{m}}$$

After calculating the success rate, the turnaround efficiency of a VM is calculated as it is also one of the SLA-based trust parameters. Therefore, it is calculated as a ratio of the estimated turnaround time supplied by a cloud provider to the actual turnaround time generated while scheduling tasks.

It is calculated using the following Equation (10).
(10)$$TE\left(n_{v}\right)=\frac{EST_{T}}{ACT_{T}}\;$$

After the calculation of the SLA-based trust parameters from Equations (8)–(10), then the trust in a cloud service provider can be calculated using Equation (11).
(11)$$trust^{csp}=X1*AV\left(n_{v}\right)+X2*SR\left(n_{v}\right)*X3*TE\left(n_{v}\right)$$

From Equation (11), $X=\left\{X1,X2,X3\right\}$ are the positive weights assigned for the calculation of trust in a cloud service provider and the X value lies between 0 and 1; these weights are calculated using the method in [32]. For X1=0.5, X2=0.2, X3=0.1, these weights will vary with respect to the requests with arrive on the cloud console. Therefore, with these weights and using Equation (11), the trust in a cloud service provider is calculated.

After the identification of the trust parameters, the makespan for the set of considered tasks is calculated as the makespan indirectly affects the quality of service which, in turn, affects the trust in a cloud provider. Therefore, we identified that there is a relationship between the makespan and SLA-based trust parameters. For calculating the makespan, initially we need to identify the execution time of a task on a corresponding VM and we posed a deadline constraint in such a way that when a request is made by a user, then immediately it should be assigned to a VM or, otherwise, it should wait until the current task has being executed on that corresponding VM. Therefore, the execution time of a task can be calculated using Equation (12).
(12)$$ex_{m_{t}}=\frac{ex_{m}}{pr_{n_{v}}}$$

After the calculation of the execution time, we calculated the finish time of the corresponding task as we posed a deadline constraint; therefore, every time a request is raised by a user, a VM needs to be assigned after the completion of the current task or assigned immediately if no tasks are currently running. Therefore, the finish time of a task needs to be identified as the deadline constraint is posed so the finish time is always less than the deadline of a task. Therefore, we calculated the finish time using Equation (13).
(13)$$ft_{m_{t}}=\;n_{v}+ex_{m_{t}}$$

To preserve trust in a cloud service provider, it is necessary to finish a task within its deadline. Therefore, the finish time of a task should be less than the deadline of task. This is calculated using Equation (14).
(14)$$ft_{m}{}_{t}\leq \;dl_{m}{}_{t}$$

Now we can calculate the makespan; this indirectly impacts on the trust in a cloud provider as it is defined as the time taken to execute a task on a particular VM. It is calculated using Equation (15).
(15)$$ma_{m_{t}}=\max\left(ft_{m_{t}}\left(n_{v}\right)\right)$$
(16)$$\min ft_{m}{}_{t}\left(m_{t}n_{v}\right)=\underset{t=1}{\sum }\underset{v=1}{\sum }\psi_{tv}ft_{m}{}_{t}\left(m_{t}n_{v}\right)$$

From Equation (16), $\psi_{tv}=1$ if a task t is assigned to a v VM; otherwise, it is to be set as 0.

Up to now, we carefully modeled the scheduler mathematically but in order to preserve trust and to optimize the parameters, we used a firefly optimization algorithm for improving the SLA-based trust parameters and the makespan. 

### 3.3. Fitness Function for Trust-Aware Task Scheduling Algorithm (TAFFA) Using Firefly Optimization

This subsection discusses the fitness function used in our proposed trust-aware task scheduling algorithm. It is calculated using the following Equation (17).
(17)$$f\left(x\right)=\psi_{1}*ma_{m_{t}}+\psi_{2}*AV\left(n_{v}\right)+\psi_{3}*SR\left(n_{v}\right)+\psi_{4}*TE\left(n_{v}\right)\;$$
where
(18)$$\psi_{1}+\psi_{2}+\psi_{3}+\psi_{4}=1$$

From Equations (17) to (18), we calculated the parameters mentioned above. When this fitness function is evaluated, the makespan needs to be minimized and the availability, success rate, and turnaround efficiency need to be improved. The next section discusses the methodology used in our task scheduler and the proposed trust-aware task scheduling algorithm using firefly optimization is discussed.

## 4. Proposed Trust-Aware Task Scheduling Algorithm Using Firefly Optimization

This section discusses the proposed trust-aware task scheduling algorithm using the firefly optimization algorithm. Initially, we will discuss the methodology used to model our scheduling algorithm. The proposed approach was modeled using firefly optimization as it requires less number of iterations and much of the problem space is traversed by fireflies because it is suitable for solving the NP-hard problem, which is the exact analogy for task scheduling in the cloud computing paradigm. The firefly optimization approach is a nature-inspired algorithm which mimics the behavior of fireflies and their attraction patterns based on the flashlight patterns discussed in [33]. The basic approach of firefly optimization and how two flies can be attracted to each other using flashlights are discussed in [34]. Given the nature of fireflies, they are more capable of solving NP-hard and multi-dimensional problems [35], which are similar to the task scheduling problems in cloud computing. Therefore, we chose the firefly optimization algorithm to solve the task scheduling problem in the cloud paradigm. These fireflies can be attracted to other flies based on the brightness of another fly. Naturally, they will be attracted towards flies of another gender. They can identify the gender of the fly based on the intensity of the flashlights, which are used for communication. Generally, in the firefly optimization approach, these cases need to be considered: (1) assuming that all flies belong to the same gender and they will attract each other irrespective of their gender. (2) A less-bright firefly will be attracted to the brighter firefly, i.e., attractiveness is directly proportional to the brightness. (3) If a fly is brighter than all the other flies in the search space, then it moves randomly. Therefore, if the flies’ light is less intense, i.e., they are less bright, then the distance between them increases as attractiveness is directly proportional to brightness. Therefore, to proceed with this approach, we need to calculate the brightness, i.e., the intensity and attractiveness. 

Initially, the intensity of a firefly can be calculated using Equation (19).
(19)$$Int\left(s\right)=\frac{Int\left(r\right)}{s^{2}}$$

Attractiveness always depends on how light is absorbing and its brightness. Therefore, the brightness intensity and light absorption coefficient are related to each other. This is indicated in Equation (20).
(20)$$Int=Int_{0}.e^{-\Omega s^{2}}$$

After calculating the intensity of the flies, we then need to identify the distance between two flies. This is calculated using Equation (21).
(21)$$s=\left|x^{i}-x^{j}\right|=\frac{1}{r}\sqrt{\overset{r}{\underset{z=1}{\sum }}{\left(x^{\left\{i,z\right\}}-x^{\left\{j,z\right\}}\right)}^{2}}$$
where *r* is the dimension of a firefly.

After the calculation of the distance, we identified the population of fireflies and dispersed them and defined the movement of a firefly i attracted towards j, which is a brighter firefly than i, for the number of iterations indicated as u. The movement of the firefly are calculated using Equation (22).
(22)$$x_{u+1}^{i}=x_{u}^{i}+\Gamma_{0}.e^{-}\left\{\frac{s^{2}}{d^{2}}\right\}\left(x_{u}^{j}-x_{u}^{i}\right)+\gamma \;EUR{\;}_{i}$$

After defining the movement of the firefly, we need to calculate the randomized damping constant, which is calculated using Equation (23).
(23)$$\gamma =\gamma_{0}.\theta_{u}$$
where θ lies in between 0 and 1. θ is a damping constant.

### Proposed Trust-Aware Task Scheduling Algorithm (TAFFA) in Cloud Computing Using Firefly Optimization Algorithm

This algorithm initially starts with a random firefly population and thereafter calculated the priorities of the incoming tasks using Equation (6) and calculated the priorities of the VMs using Equation (7). After calculating the priorities, the fitness function was evaluated using Equation (17). In next step, the intensity of the fireflies was calculated using Equations (19) and (20). In next step, the distance between the fireflies was evaluated by Equations (21) and (22). Then, if a firefly has a better brightness, then we calculated the corresponding parameters by using Equations (8)–(10) and (15), respectively. If these parameters are optimized, then we identify the firefly as the best solution and update its value as the best value or if the parameters are not optimized, we repeat this procedure for all the iterations until it arrives at the best solution. 


**Algorithm 1:** Trust-Aware Task Scheduling Algorithm (TAFFA) in Cloud Computing Using Firefly Optimization Algorithm.**Input:** set of mt=m1,m2,m3,….mt tasks, set of VMs nv=n1,n2,n3,…nv, set of hosts hoq=ho1,ho2,ho3…hoq set of datacenters dp=d1,d2,d3,…dp.**Output:** generates schedules using TAFFA while minimizing makespan, improving AV(n_v), SR(n_v), TE(n_v).Start.Initialize random firefly population.Calculate priorities of incoming tasks using Equation (6).Calculate priorities of VMs using Equation (7).Evaluate fitness function in Equation (17).for every firefly docalculate intensity using Equations (19) and (20).Calculate distance and movement of fireflies using Equation (21) and Equation (22).If a firefly has more brightness thencalculate mamt, AVnv, SRnv, TEnv using Equations (8)–(10) and (15), respectively.Return best solutionselserepeat the same procedure until best solution arrived for all iterations.End ifStop.


## 5. Simulations and Results

In this section, discussion about extensive simulations are conducted using the Cloudsim [36] simulator. This simulator presents the entire cloud environment with the support of Java programming. For conducting these simulations, in this research, we used two types of workloads, i.e., randomized workloads with different distributions fabricated as uniform, normal, left skewed, and right skewed distributions. The second type of workload considered is the real time worklogs taken from HPC2N [37] and NASA [38] computing clusters. After taking the workload from these two types, we evaluated our proposed TAFFA approach with these algorithms. After evaluating our proposed approach, we compared our proposed TAFFA with the existing ACO, GA, and PSO.

### 5.1. Settings Used for Simulation

We used Cloudsim [36] for our entire simulation. For this simulation, we used 16 GB RAM, M1 chip processor, and MAC operating system, which provides virtualization support. We used a randomized fabricated workload which is represented as b01, b02, b03, b04, b05, and b06 with different distributions. After using the randomized workload, the realtime parallel worklog archives were used from HPC2N [37] to NASA [38]. These randomized distributions are of different categories in which b01 indicates uniform distribution in which all tasks are equally distributed. b02 represents normal distribution, which represents a higher number of medium tasks and a lower number of small and large tasks. b03 represents left skewed distribution, which consists of a higher number of medium tasks and a lower number of small tasks. b04 represents right skewed distribution, which consists of a higher number of small tasks and a lower number of medium tasks. In our work, we represented HPC2N as b05 and NASA worklogs are represented as b06. We considered the simulation settings from [39].

### 5.2. Calculation of Makespan

In this work, the makespan is the evaluated makespan for the proposed TFFA scheduler and it is needed to calculate the makespan while designing a scheduler as the effectiveness of any scheduler depends on the generated makespan. The minimization of the makespan improves the performance of the scheduler. Therefore, we calculated the makespan by providing 100 to 1000 tasks and ran a simulation for 50 iterations. The configuration settings in Table 3 were used for the simulation and the given workloads of b01, b02, b03, b04, b05, and b06, respectively. The proposed TAFFA approach was compared with the baseline algorithms, i.e., ACO, GA, and PSO approaches. Table 4 indicates the generated makespan for different workloads given as the input to the TAFFA scheduler and from Figure 2, it is clearly evident that the proposed TAFFA scheduler outperforms the baseline approaches for the generated makespan for 50 iterations for the considered tasks.

### 5.3. Calculation of Availability

After calculating the makespan for the proposed TAFFA scheduler, the SLA-based trust parameters were calculated to improve the trust and to calculate the availability as it is an important parameter from both the perspectives of the cloud consumer and service provider. Improving the availability percentage benefits the cloud provider in view of the quality of service, which improves the reliability and, thereby, the trust in a cloud provider will be improved. Therefore, the availability of virtual resources is calculated by providing 100 to 1000 tasks and we ran a simulation for 50 iterations. The configuration settings in Table 3 were used for the simulation and the workloads of b01, b02, b03, b04, b05, and b06, respectively. The proposed TAFFA approach was compared with the baseline algorithms, i.e., ACO, GA, and PSO approaches. Table 5 indicates the generated availability for different workloads given as the input to the TAFFA scheduler and from Figure 3, it is clearly evident that the proposed TAFFA scheduler outperforms the baseline approaches for the generated availability for 50 iterations for the considered tasks.

### 5.4. Calculation of Success Rate

The success rate of a virtual resource improves the trust in a cloud provider indirectly as a successful execution of tasks improves the quality of service which, in turn, minimizes SLA violations and, thereby, the trust in the cloud provider increases. Therefore, the success rate of virtual resources was calculated by giving 100 to 1000 tasks and we ran a simulation for 50 iterations. The configuration settings in Table 3 were used for the simulation and the given workloads of b01, b02, b03, b04, b05, and b06, respectively. The proposed TAFFA was compared with the baseline algorithms, i.e., ACO, GA, and PSO approaches. Table 6 indicates the generated success rate for the different workloads given as the input to the TAFFA scheduler and from Figure 4, it is clearly evident that the proposed TAFFA scheduler outperforms the baseline approaches for the generated success rate for 50 iterations for the considered tasks.

### 5.5. Calculation of Turnaround Efficiency

The turnaround efficiency of virtual resources impacts the quality of service as it relates to the successful execution of tasks and the time taken to execute and respond to the user for their corresponding requests. When the turnaround efficiency improves, this in turn improves the quality of service; thereby, the trust in the cloud provider increases. Therefore, the turnaround efficiency of the virtual resources was calculated by providing 100 to 1000 tasks and we ran a simulation for 50 iterations. The configuration settings in Table 3 were used for the simulation and the given workloads of b01, b02, b03, b04, b05, and b06, respectively. The proposed TAFFA approach is compared with the baseline algorithms, i.e., ACO, GA, and PSO approaches. Table 7 indicates the generated turnaround efficiency for different workloads given as the input to the TAFFA scheduler and from Figure 5, it is clearly evident that the proposed TAFFA scheduler outperforms the baseline approaches for the generated turnaround efficiency for 50 iterations for the considered tasks.

### 5.6. Discussion of Results

This subsection discusses the analysis of the simulated results and the improvement of the parameters which were mentioned in Section 5.2, Section 5.3, Section 5.4, Section 5.5 and Section 5.6. We already mentioned that we used Cloudsim [35] for extensive simulations. Our entire simulations were carried out with different datasets which were fabricated and with real time worklogs and they are mentioned as b01, b02, b03, b04, b05, and b06, respectively. We compared our proposed TAFFA scheduler with the existing baseline algorithms, i.e., ACO, GA, and PSO, by running a simulation with 50 iterations. We already presented the results for our measured parameters in the above Section 5.2, Section 5.3, Section 5.4 and Section 5.5. From the results, we draw inferences in such a way that measure how far our proposed TAFFA improved the parameters and generated schedules effectively. We presented the improvement of the makespan over the baseline approaches in Table 8 for the different workloads we used; in Table 9, we presented the improvement of the availability over the baseline approaches for the different workloads; in Table 10, we presented the improvement of the success rate over the baseline approaches for the different workloads; and in Table 11, we presented the improvement of the turnaround efficiency over the baseline approaches for the different workloads. Finally, from all these results, we can say that our proposed TAFFA scheduler improves the basic parameters, i.e., the makespan, energy consumption, as well as the SLA-based trust parameters, thereby improving the quality of service and trust in the cloud provider for the satisfaction of the cloud user.

#### 5.6.1. Improvement of Makespan

This subsection discusses the improvement of the makespan as the effectiveness of the task scheduler primarily relies on the makespan. Therefore, the minimization of the makespan is to be considered as the primary aspect. The proposed TAFFA was evaluated over the baseline approaches, i.e., ACO, GA, and PSO. The reason we compared the proposed TAFFA over these algorithms is that many earlier authors modeled task schedulers with these approaches, but with the as-discussed disadvantages of ACO, GA, and PSO algorithms in the related works and different workloads, i.e., different statistical distributions and realtime worklogs, we considered these for the simulation. In Table 8, the proposed TAFFA was minimized over ACO for the given workloads b01, b02, b03, b04, b05, and b06 with 25.09%, 27.73%, 25.49%, 11%, 22.55%, and 27.45%, respectively. It was minimized over the GA for the given workloads of b01, b02, b03, b04, b05, and b06 with 37.6%, 26.62%, 30.1%, 19.86%, 32.08%, and 35.75%, respectively. It was minimized over PSO for the given workloads of b01, b02, b03, b04, b05, and b06 with 32.38%, 16.92%, 25.65%, 16.74%, 33.48%, and 23.11%, respectively. Therefore, the proposed TAFFA greatly minimizes the makespan over the mentioned algorithms in Table 8.

#### 5.6.2. Improvement of Availability

This subsection discusses the improvement of the availability as trust in a cloud provider is based on the availability of virtual resources in the cloud paradigm. Therefore, the improvement of the availability of virtual resources is related to the quality of service provided by a cloud paradigm. The degradation of the availability leads to damaging the quality of service, which in turn affects the trust in a cloud paradigm. Therefore, the availability of virtual resources is calculated carefully using the proposed TAFFA. Different statistical distributions and realtime worklogs are considered for the simulation. In Table 9, the proposed TAFFA was improved over ACO for the given workloads of b01, b02, b03, b04, b05, and b06 with 27.28%, 27.4%, 34.64%, 42.98%, 53.56%, and 58.49%, respectively. It is improved over the GA for the given workloads of b01, b02, b03, b04, b05, and b06 with 19.54%, 15.32%, 17.93%, 34.82%, 49.96%, and 55.29%, respectively. It is improved over PSO for the given workloads of b01, b02, b03, b04, b05, and b06 with 22.4%, 19.02%, 24.47%, 22.87%, 29.62%, and 30.25%, respectively. Therefore, the proposed TAFFA greatly improves the availability over the mentioned algorithms in Table 9.

#### 5.6.3. Improvement of Success Rate

This subsection discusses the improvement of the success rate trust in a cloud provider is based on the success rate of virtual resources in cloud paradigm. Therefore, the improvement of the success rate of virtual resources is related to the quality of service provided by a cloud paradigm. The success rate of tasks highly impacts the quality of service provided by a cloud provider, which in turn relates to trust in a cloud provider. Therefore, the success rate of virtual resources is calculated carefully using the proposed TAFFA. Different statistical distributions and realtime worklogs are considered for the simulation. In Table 10, the proposed TAFFA improved over ACO for given workloads of b01, b02, b03, b04, b05, and b06 with 30.48%, 66.7%, 53.56%, 56.82%, 60.08%, and 65.14%, respectively. It is improved over the GA for the given workloads of b01, b02, b03, b04, b05, and b06 with 24.84%, 53.79%, 58.11%, 40.23%, 48.43%, and 72.36%, respectively. It is improved over the PSO for the given workloads of b01, b02, b03, b04, b05, and b06 with 24.86%, 33.42%, 38.89%, 49.48%, 33.78%, and 55.55%, respectively. Therefore, the proposed TAFFA greatly improves the success rate over the mentioned algorithms in Table 10.

#### 5.6.4. Improvement of Turnaround Efficiency

This subsection discusses the improvement of the turnaround efficiency as trust in a cloud provider is based on the turnaround efficiency of the virtual resources in a cloud paradigm. Therefore, the improvement of the turnaround efficiency of tasks on virtual resources is related to the quality of service provided by a cloud paradigm. The turnaround efficiency of tasks on virtual resources highly impacts the quality of service of a cloud provider, which in turn impacts the trust in a cloud provider. Therefore, the turnaround efficiency of tasks on virtual resources are calculated carefully using the proposed TAFFA. Different statistical distributions and realtime worklogs were considered for the simulation. In Table 11, the proposed TAFFA improved over ACO for the given workloads b01, b02, b03, b04, b05, and b06 with 88.75%, 59.21%, 38.92%, 41.85%, 82.38%, and 37.8%, respectively. It is improved over the GA for the given workloads of b01, b02, b03, b04, b05, and b06 with 71.57%, 51.91%, 36.24%, 26.77%, 73.66%, and 51.41%, respectively. It is improved over PSO for the given workloads of b01, b02, b03, b04, b05, and b06 with 67.62%, 38.33%, 29.77%, 26.55%, 41.04%, and 33.18%, respectively. Therefore, the proposed TAFFA greatly improves the turnaround efficiency over the mentioned algorithms in Table 11.

## 6. Conclusions and Future Work

Task scheduling is a challenging aspect in a cloud computing paradigm as workloads in the cloud paradigm as incoming workloads onto the cloud console arise from various heterogeneous resources with different types and different runtime processing capacities. Therefore it is difficult for a cloud provider to schedule these tasks onto the appropriate virtual resources. The ineffective mapping of tasks onto the virtual resources leads to the degradation of the quality of service and results in an increase in the makespan and which, in turn, leads to a violation of the SLA between the cloud provider and the user, which damages the trust in the cloud provider. Therefore, in order to address the SLA-based trust parameters, we carried out research in which the priorities of all incoming tasks and VMs are calculated and then precisely mapped the tasks onto the VMs. We chos the firefly optimization approach to model our trust-aware task scheduler (TAFFA) and the entire simulations were carried out on Cloudsim. We carried out extensive simulations by generating tasks using different randomized task distributions and to check the efficacy of our approach, we used the standard benchmark worklogs of HPC2N and NASA computing clusters. Finally, our proposed TAFFA was evaluated over the state-of-the-art algorithms and from the results, it evident that TAFFA shows its dominance over the existing approaches by minimizing the makespan and improving the avaialability, success rate, and turnaround efficiency. Our proposed TAFFA approch has some limitations as the tasks are generated from various heterogeneous resources and the proposed TAFFA is not able to predict upcoming tasks from various users. In the future, we intend to develop our task scheduling algorithm by using an artificial intelligence approach.

## Figures and Tables

**Figure 1 sensors-23-01384-f001:**
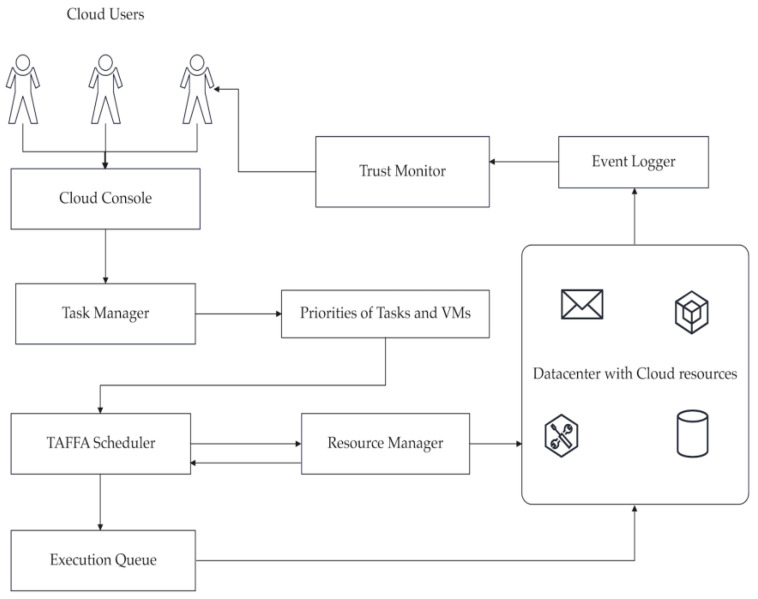
Proposed system architecture.

**Figure 2 sensors-23-01384-f002:**
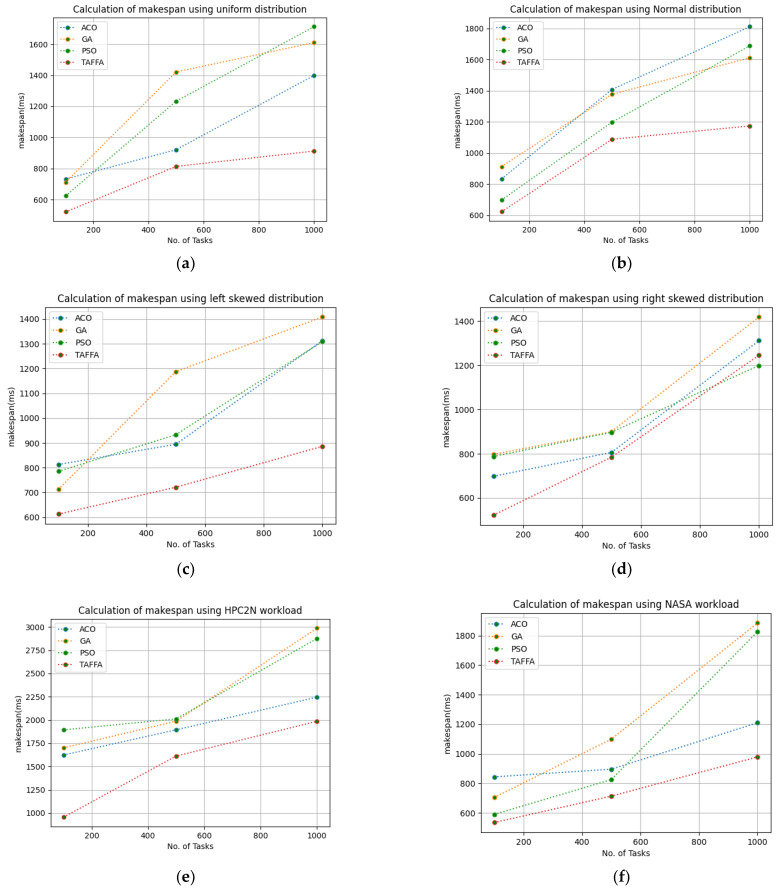
Calculation of makespan using various workloads: (**a**) calculation of makespan using uniform distribution; (**b**) calculation of makespan using normal distribution; (**c**) calculation of makespan using left skewed distribution; (**d**) calculation of makespan using right skewed distribution; (**e**) calculation of makespan using HPC2N Workload; (**f**) calculation of makespan using NASA Workload.

**Figure 3 sensors-23-01384-f003:**
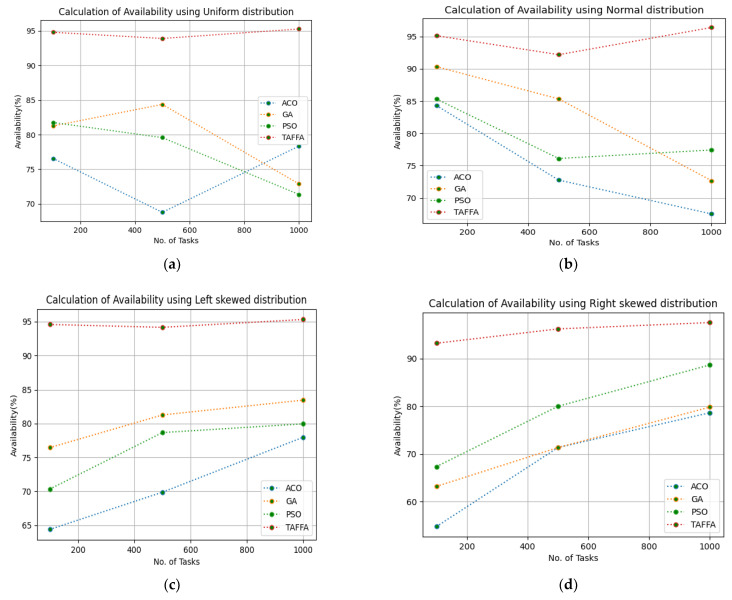
Calculation of availability using various workloads: (**a**) calculation of availability using uniform distribution; (**b**) calculation of availability using normal distribution; (**c**) calculation of availability using left skewed distribution; (**d**) calculation of availability using right skewed distribution; (**e**) calculation of availability using HPC2N workload; (**f**) calculation of availability using NASA workload.

**Figure 4 sensors-23-01384-f004:**
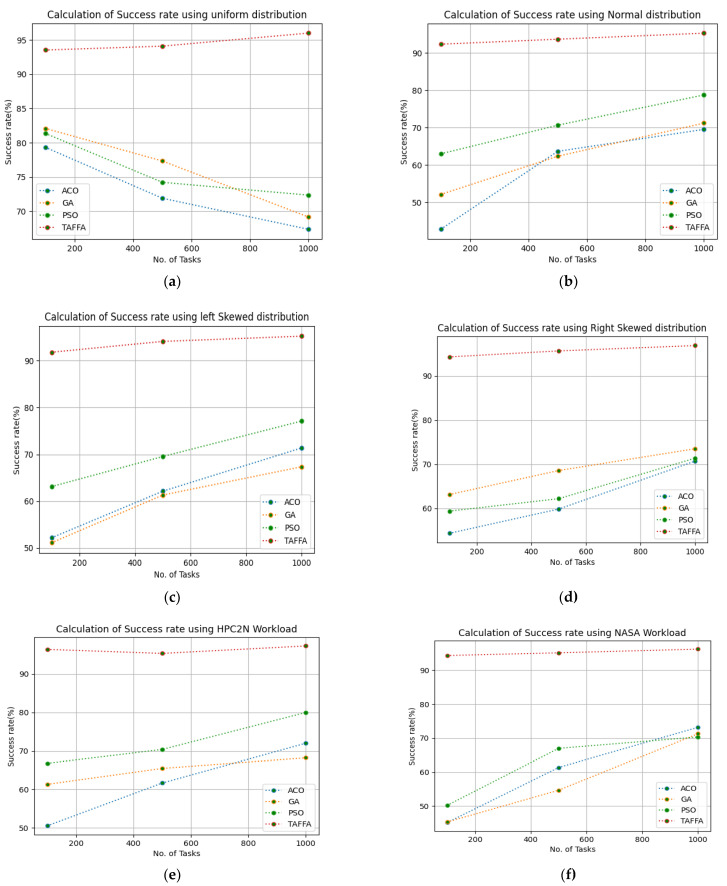
Calculation of success rate using various workloads: (**a**) calculation of success rate using uniform distribution; (**b**) calculation of success rate using normal distribution; (**c**) calculation of Success rate using left skewed distribution; (**d**) calculation of success rate using right skewed distribution; (**e**) calculation of success rate using HPC2N workload; (**f**) calculation of success rate using NASA workload.

**Figure 5 sensors-23-01384-f005:**
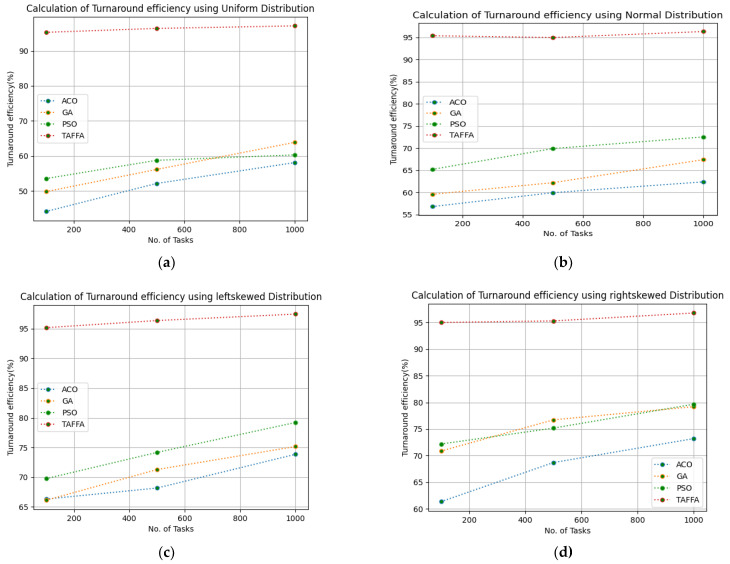
Calculation of turnaround efficiency using various workloads: (**a**) calculation of turnaround efficiency using uniform distribution; (**b**) calculation of turnaround efficiency using normal distribution; (**c**) calculation of turnaround efficiency using left skewed distribution; (**d**) calculation of turnaround efficiency using right skewed distribution; (**e**) calculation of turnaround efficiency using HPC2N workload; (**f**) calculation of turnaround efficiency using NASA workload.

**Table 1 sensors-23-01384-t001:** Summary of various task scheduling algorithms in cloud computing and addressed parameters.

Authors	Technique Used	Parameters
[6]	Adaptive PSO	Throughput, makespan, resource utilization.
[7]	PSO	Makespan, energy consumption.
[8]	PSO	Resource utilization, makespan, task rejection, penalty cost, total cost.
[9]	CSO	Makespan, total power cost, migration time, energy consumption.
[10]	LW-PSO	Execution time, turnaround time, response time.
[11]	Ba-PSO	Total execution time, wait time, scalability, load balancing of tasks.
[12]	OPSO	Makespan, energy consumption.
[13]	EMVO	Makespan, resource utilization.
[14]	GA-FPA	Resource utilization, computation cost, completion time.
[15]	EPETS	Makespan, energy Consumption.
[16]	HSGA	Makespan, cost, response time.
[17]	MFTGA	Fault tolerance, execution time, cost, energy consumption, SLA violation.
[18]	GA-WOA	Throughput, processing cost, computation cost, processing time.
[19]	GAECS	Makespan, energy consumption.
[20]	GSAGA	Processing capacity, energy consumption.
[21]	FPSO-GA	Load balancing of tasks.
[22]	MOTS-ACO	Makespan, turnaround time, load balancing, power efficiency.
[23]	AC-PSO	Makespan, total cost, resource utilization.
[24]	R-ACO	Makespan.
[25]	MR-LBA	Execution time and execution cost.
[26]	ACOBF	Energy Consumption, makespan, resource utilization.
[27]	Adaptive ACO	Task completion time, execution cost, load balance of tasks.
[28]	HEFT-ACO	Makespan, cost.
[29]	JRA-ACO-GA	Resource utilization.
[30]	ACO-NN	Load balancing.
[31]	DGWO	Makespan, computation and transmission costs.

**Table 2 sensors-23-01384-t002:** Notations used in system architecture.

Entity	Meaning
$m_{t}$	Set of tasks.
$n_{v}$	Set of VMs.
$d_{p}$	Set of datacenters.
$ho_{q}$	Set of hosts.
$load_{n}{}_{v}$	workload on virtual resources.
$load_{ho_{q}}$	Workload on physical hosts.
$pr_{n}{}_{v}$	Processing capacity of virtual resources.
$prio_{m_{t}}$	Priorities of considered tasks
$prio_{n}{}_{v}$	Priorities of VMs based on electricity unit cost.
$ma_{m_{t}}$	Makespan.
$dl_{m_{t}}$	Deadline constraint.
$ex_{m_{t}}$	Execution time.
$ft_{m_{t}}$	Finish time.
$AV\left(n_{v}\right)$	Availability of VMs.
$SR\left(n_{v}\right)$	Success rate of VMs.
$TE\left(n_{v}\right)$	Turnaround efficiency.
$trust^{csp}$	Trust in a cloud provider.

**Table 3 sensors-23-01384-t003:** Configuration settings for simulation.

Name	Quantity
No. of tasks considered	100–1000
Length of tasks	900,000
Memory of host	16 GB
Storage capacity of host	2 TB
Network bandwidth capacity	200 Mbps
No. of VMs	50
Memory capacity of VM	2 GB
Bandwidth of VM	20 Mbps
Processing elements	1100 MIPS
Hypervisor used	Xen
Hypervisor type	Monolithic
Operating system	MAC
Datacenters used	10

**Table 4 sensors-23-01384-t004:** Calculation of makespan using various workloads.

Algorithm												
No. of Tasks	ACO			GA			PSO			TAFFA		
	$\vec{q}$	δ	Best	$\vec{q}$	δ	Best	$\vec{q}$	δ	Best	$\vec{q}$	δ	Best
b01												
100	752.8	1.32	732.8	767.99	1.99	712.24	675.6	2.43	623.88	545.87	2.67	521.34
500	941.3	2.88	921.2	1487.32	2.56	1421.12	1267.34	1.57	1232.87	853.21	1.78	813.67
1000	1421.6	0.89	1398.8	1656.12	1.12	1610.33	1763.3	2.44	1712.45	945.32	0.98	912.57
b02												
100	857.21	1.78	834.12	857.21	989.34	2.45	912.87	753.58	1.21	698.35	678.12	1.67
500	1456.3	2.67	1408.34	1456.3	1425.78	2.89	1377.34	1265.21	0.69	1198.32	1146.66	1.11
1000	1854.21	0.34	1812.22	1854.21	1688.6	1.43	1612.66	1728.76	0.87	1689.34	1254.65	1.78
b03												
100	853.7	1.76	812.9	753.21	2.12	712.35	834.12	1.28	785.23	653.54	0.78	612.56
500	987.3	1.98	894.67	1287.65	2.58	1187.32	986.12	0.34	933.12	788.12	0.56	721.12
1000	1378.6	2.45	1312.5	1457.32	1.21	1408.35	1398.5	1.97	1309.34	924.58	0.17	886.56
b04												
100	745.32	1.23	697.86	853.57	2.16	797.36	876.14	2.12	787.56	554.12	1.77	521.45
500	843.24	1.58	806.35	956.35	1.32	899.79	987.12	2.56	897.13	824.78	2.12	784.56
1000	1387.54	2.16	1311.32	1488.43	1.76	1418.36	1247.98	2.25	1198.72	1378.9	2.74	1245.34
b05												
100	1687.3	2.87	1623.45	1745.5	1.87	1699.9	1931.2	2.14	1893.5	988.46	1.13	954.12
500	1923.4	3.34	1894.3	2267.6	1.38	1987.34	2178.5	3.12	2012.45	1688.12	1.89	1612.34
1000	2756.3	2.18	2245.8	3088.5	0.34	2987.45	2923.43	1.59	2877.12	2068.7	1.35	1986.21
b06												
100	887.35	1.78	843.21	757.65	1.27	703.56	634.21	1.29	589.17	578.12	1.47	534.12
500	956.88	1.12	895.34	1189.26	2.12	1098.23	894.32	1.87	824.66	798.12	1.34	712.67
1000	1278.9	0.78	1209.56	1998.32	1.09	1885.12	1856.23	1.67	1826.77	1098.7	2.13	978.55

**Table 5 sensors-23-01384-t005:** Calculation of availability using various workloads.

Algorithm												
No. of Tasks	ACO			GA			PSO			TAFFA		
	$\vec{q}$	δ	Best	$\vec{q}$	δ	Best	$\vec{q}$	δ	Best	$\vec{q}$	δ	Best
b01												
100	82.77	1.12	76.57	93.4	0.68	81.25	89.67	1.77	81.76	97.88	0.87	94.78
500	74.36	2.56	68.78	87.99	0.87	84.37	84.32	1.36	79.57	95.43	0.36	93.89
1000	82.18	1.09	78.35	78.97	0.36	72.88	77.56	1.23	71.36	96.57	0.78	95.26
b02												
100	88.76	0.98	84.31	92.64	0.79	90.32	88.98	1.12	85.32	96.76	0.87	95.12
500	79.88	1.12	72.77	89.57	0.43	85.34	78.16	1.29	76.11	95.77	2.78	92.18
1000	69.89	1.04	67.53	77.84	0.43	72.67	82.18	1.99	77.43	97.89	0.57	96.39
b03												
100	69.38	1.88	64.38	79.78	2.56	76.45	74.58	1.28	70.33	96.21	0.77	94.59
500	76.88	1.54	69.87	84.78	0.85	81.27	81.38	2.57	78.67	95.33	0.87	94.17
1000	79.58	1.29	77.99	85.22	1.77	83.46	85.31	0.87	79.96	97.55	0.71	95.33
b04												
100	58.99	1.76	54.78	67.37	1.93	63.21	69.88	1.45	67.36	94.54	0.52	93.21
500	77.29	2.78	71.37	79.09	1.57	71.36	82.55	1.11	80.02	97.11	0.54	96.22
1000	82.46	1.84	78.67	81.37	1.42	79.86	91.24	3.75	88.67	98.31	0.26	97.55
b05												
100	57.86	1.65	53.47	64.99	1.12	59.33	76.39	2.90	73.24	94.39	0.72	92.66
500	64.53	1.63	63.53	68.36	2.37	62.19	78.11	3.08	67.66	97.13	0.63	95.14
1000	73.32	1.54	70.38	74.47	3.12	68.84	83.31	2.13	79.57	98.41	0.35	96.88
b06												
100	55.33	1.15	51.76	58.18	0.34	51.57	71.8	2.21	68.99	95.31	0.25	92.16
500	63.32	2.98	59.44	66.11	2.14	59.88	74.32	1.87	71.56	96.18	0.91	94.55
1000	72.31	2.46	69.46	79.57	2.86	74.33	78.11	1.12	76.85	98.42	0.41	96.11

**Table 6 sensors-23-01384-t006:** Calculation of success rate using various workloads.

Algorithm												
No. of Tasks	ACO			GA			PSO			TAFFA		
	$\vec{q}$	δ	Best	$\vec{q}$	δ	Best	$\vec{q}$	δ	Best	$\vec{q}$	δ	Best
b01												
100	84.55	2.83	79.33	86.11	0.55	82.09	82.17	1.38	81.36	96.31	0.61	93.54
500	74.24	1.32	71.88	79.24	1.53	77.33	79.33	1.21	74.21	95.77	0.59	94.11
1000	68.11	1.82	67.33	72.11	0.75	69.12	76.41	0.92	72.33	97.12	0.21	96.02
b02												
100	45.87	1.54	42.77	56.12	1.38	52.11	64.77	1.76	62.99	97.35	0.21	92.37
500	68.33	2.11	63.66	65.21	2.17	62.35	78.46	2.82	70.66	96.78	0.43	93.71
1000	72.11	1.82	69.54	78.17	1.66	71.22	82.87	1.12	78.77	97.69	0.19	95.32
b03												
100	55.16	1.54	52.18	57.34	2.86	51.12	68.33	1.53	63.12	93.31	0.57	91.78
500	68.33	2.26	62.13	66.15	1.08	61.31	73.54	2.57	69.54	96.17	0.21	94.09
1000	77.35	2.67	71.39	72.11	1.73	67.36	82.12	2.09	77.13	97.65	0.56	95.21
b04												
100	58.56	0.57	54.33	67.16	2.57	63.12	62.77	1.29	59.36	97.96	1.24	94.31
500	61.35	2.11	59.83	71.88	1.88	68.57	69.19	1.38	62.18	98.12	2.12	95.67
1000	73.12	1.67	70.72	78.24	2.36	73.52	78.33	2.89	71.38	98.89	2.56	96.88
b05												
100	54.88	1.77	50.56	64.22	1.87	61.33	69.86	1.37	66.78	97.87	0.78	96.36
500	64.78	1.88	61.67	69.88	1.34	65.46	73.55	2.87	70.37	96.79	0.57	95.32
1000	77.47	1.93	71.99	72.17	2.09	68.21	81.26	1.77	79.98	98.81	1.16	97.26
b06												
100	46.77	1.77	45.14	48.12	1.31	45.31	53.52	1.21	50.21	95.88	0.92	94.32
500	62.77	0.87	61.36	59.71	2.76	54.68	68.31	1.32	66.98	97.67	0.37	95.12
1000	78.33	1.32	73.17	74.16	1.88	71.27	72.31	1.10	70.32	98.32	0.56	96.21

**Table 7 sensors-23-01384-t007:** Calculation of turnaround efficiency using various workloads.

Algorithm												
No. of Tasks	ACO			GA			PSO			TAFFA		
	$\vec{q}$	δ	Best	$\vec{q}$	δ	Best	$\vec{q}$	δ	Best	$\vec{q}$	δ	Best
b01												
100	48.42	1.33	44.17	51.13	1.48	49.83	58.72	1.86	53.56	97.28	0.57	95.27
500	56.12	0.97	52.17	59.35	1.71	56.17	60.36	1.37	58.77	96.48	0.26	96.39
1000	62.77	2.18	58.53	66.13	2.82	63.91	63.67	1.37	60.32	98.17	0.92	97.11
b02												
100	53.31	1.22	56.78	61.82	1.64	59.56	68.37	1.88	65.18	97.32	1.88	95.36
500	61.87	2.56	59.92	68.33	1.24	62.18	71.37	1.35	69.88	96.75	2.21	94.96
1000	68.99	1.36	62.37	69.98	1.72	67.38	76.67	1.41	72.51	95.77	1.12	96.31
b03												
100	68.76	1.09	66.32	69.17	1.25	66.12	72.86	2.53	69.76	96.31	1.47	95.17
500	70.35	1.11	68.18	74.85	1.63	71.29	75.47	1.65	74.18	98.48	1.28	96.37
1000	75.21	0.98	73.87	79.63	1.42	75.18	82.57	1.87	79.23	98.76	0.76	97.46
b04												
100	64.65	1.21	61.37	74.17	2.47	70.87	77.67	1.62	72.19	95.43	0.12	94.98
500	71.77	2.57	68.71	79.37	1.57	76.73	79.37	2.15	75.18	96.19	0.32	95.27
1000	77.31	1.75	73.21	83.17	2.53	79.21	82.56	2.37	79.67	98.47	0.56	96.76
b05												
100	49.67	1.29	47.38	53.26	0.27	51.37	65.78	1.67	63.67	96.87	0.46	95.21
500	52.77	1.21	51.21	58.54	0.78	57.31	69.11	1.79	68.66	96.38	0.21	95.55
1000	64.36	1.62	61.66	60.67	1.07	58.26	76.88	1.89	73.21	99.08	0.21	98.43
b06												
100	66.88	1.74	64.92	62.27	2.59	58.65	70.39	1.21	69.87	97.57	0.68	97.01
500	73.53	1.36	71.38	67.18	2.16	63.19	75.77	1.95	71.48	96.98	0.54	96.22
1000	78.28	1.73	76.49	75.38	1.53	72.36	80.84	1.82	78.37	99.15	0.26	98.82

**Table 8 sensors-23-01384-t008:** Improvement in makespan over existing algorithms.

Algorithms			
Improvement of Makespan			
	ACO	GA	PSO
b01	25.09%	37.6%	32.38%
b02	27.73%	26.62%	16.92%
b03	25.49%	30.1%	25.65%
b04	11%	19.86%	16.74%
b05	22.55%	32.08%	33.48%
b06	27.45%	35.75%	23.11%

**Table 9 sensors-23-01384-t009:** Improvement in availability over existing algorithms.

Algorithms			
Improvement of Availability			
	ACO	GA	PSO
b01	27.28%	19.54%	22.4%
b02	27.4%	15.32%	19.02%
b03	34.64%	17.93%	24.47%
b04	42.98%	34.82%	22.87%
b05	53.56%	49.96%	29.62%
b06	58.49%	55.29%	30.25%

**Table 10 sensors-23-01384-t010:** Improvement in success rate over existing algorithms.

Algorithms			
Improvement of Success Rate			
	ACO	GA	PSO
b01	30.48%	24.84%	24.86%
b02	66.7%	53.79%	33.42%
b03	53.56%	58.11%	38.89%
b04	56.82%	40.23%	49.48%
b05	60.08%	48.43%	33.78%
b06	65.14%	72.36%	55.55%

**Table 11 sensors-23-01384-t011:** Improvement in turnaround efficiency over existing algorithms.

Algorithms			
Improvement of Turnaround Efficiency			
	ACO	GA	PSO
b01	88.75%	71.57%	67.62%
b02	59.21%	51.91%	38.33%
b03	38.92%	36.24%	29.77%
b04	41.85%	26.77%	26.55%
b05	82.38%	73.66%	41.04%
b06	37.8%	51.41%	33.18%

## Data Availability

Not applicable.

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
