# Peer review of "An Efficient Trust-Aware Task Scheduling Algorithm in Cloud Computing Using Firefly Optimization"

_sensors, 2023, doi:10.3390/s23031384_

Round 1

Reviewer 1 Report

Defining abbreviations before using them in the abstract is a wise idea. The paper needs extensive grammatical corrections and sentence restructuring.  

There is a lack of clarity in the results and discussion. In spite of the fact that the authors have generated sufficient results, their discussion is very weak. It is important to critically analyze the results of the study in order to improve the paper's quality. It is important to present the results in a smooth manner, by explaining the graphs and tables, and also by including a why factor in the paper. Would you like to know why the results are the way they are? What can be done to improve them.  

Table 1 summarizes the parameters used, limitations, and strengths of each paper that would have enhanced its value. It may be possible for the authors to include limitations in the text of the paper. 

In addition, we, our have been frequently used in the paper, so avoid using them. 

Author Response

Reviewer-1 

Query1: Defining abbreviations before using them in the abstract is a wise idea. The paper needs extensive grammatical corrections and sentence restructuring.  

Answer: Thank you for the reviewer for their valuable suggestion. We have added appropriate abbreviations in the keywords. Abbreviations are highlighted in yellow colour. Thank you for suggestion of the reviewer and we updated manuscript with grammatical corrections and sentence restructuring.

Query2:  There is a lack of clarity in the results and discussion. Inspite of the fact that the authors have generated sufficient results, their discussion is very weak. It is important to critically analyse the results of the study in order to improve the paper's quality. It is important to present the results in a smooth manner, by explaining the graphs and tables, and also by including a why factor in the paper. Would you like to know why the results are the way they are? What can be done to improve them. 

Answer: Thank you for rising the query by the reviewer. We have clearly improved our discussion of results and parameters in the section 5 and its subsections. We have analysed the results and calculated improvement of parameters  by explaining them and why we consider those parameters are clearly explained in the section 5 in updated manuscript and changes are highlighted in yellow colour in the revised manuscript.

Query3: Table 1 summarizes the parameters used, limitations, and strengths of each paper that would have enhanced its value. It may be possible for the authors to include limitations in the text of the paper. 

Answer: Thank you for suggestion of the reviewer. Yes table 1 summarizes parameters, techniques used in existing literature review. We have included a summary of parameters which are not addressed by existing authors and limitations of existing algorithms at the end of table1 and highlighted in yellow colour.

Query4: In addition, we, our have been frequently used in the paper, so avoid using them. 

Answer: Thank you for suggestion of reviewer. We removed we, our in manuscript.

Reviewer 2 Report

In this paper, a a task scheduling algorithm using firefly optimization to minimize makespan and improve availability, success rate and turnaround efficiency. The literature review was comprehensive. The research has great application values. 

However, a few issues should be considered to make the paper clearer and easy understood.

1. Justification of firefly optimization. The selection and application of firefly optimization should be justified.

2. Right use of Acronyms, quite a few acronyms, such as SLA, AOC and PSO were used without proper definition. The full terms of these acronyms should be given when they are used first time. This will improve the readability and reduce ambiguity. 

Author Response

Reviewer-2

In this paper, a task scheduling algorithm using firefly optimization to minimize makespan and improve availability, success rate and turnaround efficiency. The literature review was comprehensive. The research has great application values. 

However, a few issues should be considered to make the paper clearer and easy understood.

Query 1:Justification of firefly optimization. The selection and application of firefly optimization should be justified.

Answer: Thank you for the Reviewer for rising this query. Yes, we have used Firefly optimization to model our task scheduler as firefly requires very less number of iterations and it is suitable for task scheduling environment in cloud computing as it is a NP hard problem. It is highlighted in yellow colour in Section 4.

Query 2: Right use of Acronyms, quite a few acronyms, such as SLA, ACO and PSO were used without proper definition. The full terms of these acronyms should be given when they are used first time. This will improve the readability and reduce ambiguity. 

Answer: Thank you for the reviewer for giving the appropriate suggestion and we have updated manuscript by keeping these acronyms in keywords for better readability and to avoid ambiguity and they are highlighted in yellow colour in keywords section.

Reviewer 3 Report

- In the abstract, change "we proposed an efficient task" to "we propose an efficient task"

- In the abstract, change "This scheduling algorithm modeled by using firefly optimization." to "This scheduling algorithm is modeled using the firefly optimization algorithm."

-change the title of Section 2 to "Related Work"

- You should review and compare the following important related papers in the literature review section:

Distributed grey wolf optimizer for scheduling of workflow applications in cloud environments

- The adverb "where" after equation 6 should be written with small letters

- Use the abbreviation Eq. for equations instead of eqn.

- The limitations of the proposed algorithm should be mentioned in the conclusion section

Author Response

Reviewer-3

Query 1: In the abstract, change "we proposed an efficient task" to "we propose an efficient task"

Answer: Thank you for the suggestion of the reviewer and we have updated the mentioned sentence in the query as per the suggestion of the reviewer and highlighted in yellow colour.

Query2: In the abstract, change "This scheduling algorithm modelled by using firefly optimization." to "This scheduling algorithm is modelled using the firefly optimization algorithm."

Answer: Thank you for the suggestion of the reviewer and we have updated the mentioned sentence in the query as per the suggestion of the reviewer and highlighted in Yellow colour.

Query3: change the title of Section 2 to "Related Work"

Answer: Thank you for the suggestion of the reviewer and we have changed the title as “Related Work” and it is highlighted in yellow colour.

Query4: You should review and compare the following important related papers in the literature review section.

Answer: Thank you for the suggestion of the reviewer and we have cited the reference mentioned by reviewer and it is highlighted in Table1. We used the same reference in Related Work and it is highlighted in yellow colour. The following reference mentioned by the reviewer is used in our manuscript.

Abed-Alguni, Bilal H., and Noor Aldeen Alawad. "Distributed Grey Wolf Optimizer for scheduling of workflow applications in cloud environments." Applied Soft Computing 102 (2021): 107113.

Query5:  The adverb "where" after equation 6 should be written with small letters

Answer:  Thank you for suggestion of the reviewer we have updated adverb “where” with small letters and it is highlighted in yellow color in the manuscript.

Query6: Use the abbreviation Eq. for equations instead of eqn.

Answer: Thank you for suggestion of the reviewer and we have updated abbreviation Eq. for equations in the manuscript and it is highlighted in yellow colour in the manuscript.

Query7: The limitations of the proposed algorithm should be mentioned in the conclusion section

Answer: Thank you for suggestion of the reviewer and as per the suggestion we added the limitations of the proposed algorithm in conclusion section.

Round 2

Reviewer 1 Report

The authors have addressed the comments but some repeated sentences have been included in the paper which authors may re-write before publication.  

line 550 to 554, 586 to 590, 618 to 623 are almost same words so author may change it either re-write or remove the duplication.  Other than that paper has been improved compared to initial version.

Author Response

The authors have addressed the comments but some repeated sentences have been included in the paper which authors may re-write before publication. 

line 550 to 554, 586 to 590, 618 to 623 are almost same words so author may change it either re-write or remove the duplication.  Other than that paper has been improved compared to initial version.

Answer: Thank you for the suggestion  of the reviewer and we have changed the sentences in updated manuscript and updated changes are highlighted in Green Colour.